# Explainable Pathfinding for Inscrutable Planners with Inductive Logic Programming

**Forest Agostinelli,**[1] **Rojina Panta,** [1] **Vedant Khandelwal** [1]**, Biplav Srivastava** [1]**, Bharath Muppasani** [1]**, Kausik Lakkaraju**[1]**, Dezhi Wu** [2]**,**

[1] AI Institute, University of South Carolina, Columbia, South Carolina, USA
[2] Department of Integrated Information Technology, University of South Carolina, Columbia, South Carolina, USA
foresta@cse.sc.edu, rpanta@email.sc.edu, vedant@mailbox.sc.edu, biplav.s@sc.edu, bharath@email.sc.edu,
kausik@email.sc.edu, dezhiwu@cec.sc.edu

## Abstract

The complexity of the solutions that artificial intelligence can learn to solve problems currently surpasses its ability to explain these solutions. In many domains, explainable solutions are a necessary condition while optimality is not. Therefore, we seek to constrain solutions to the space of solutions that can be explained to a human. To do this, we build on inductive logic programming (ILP) techniques that allow us to define robust background knowledge and inductive biases. By combining ILP with a given inscrutable planner, we are able to construct an explainable graph representing solutions to all states in the state space. This graph can then be summarized using a variety of methods such as hierarchical representations and simple if/else rules. We test our approach on Towers of Hanoi and discuss future work for applications to the Rubik's cube.

## Introduction

As artificial intelligence (AI) continues to solve problems that humans struggle to solve, there is an emerging need for humans to understand these solutions so that we can trust AI, create new educational opportunities, and even discover new knowledge. Many of these problems are pathfinding problems. That is, the problem is to find a sequence of actions (a path) to go from any given state to a goal state. Examples of AI being successfully applied to pathfinding problems include puzzles (McAleer et al. 2019; Agostinelli et al. 2019), quantum compiling (Zhang et al. 2020), chemical synthesis (Chen et al. 2020), theorem proving (Bansal et al. 2019), and program synthesis (Ellis et al. 2021). Many explainable AI (XAI) approaches for explaining how to solve pathfinding problems and other sequential decision making problems focus on explaining a plan to solve a single instance of a problem (Sreedharan et al. 2022). However, we focus on finding an explanation for solving *all* possible instances of a problem.

Depending on the application, the simplicity of an explanation can matter more than the efficiency of a solution. However, modern AI techniques, such as deep reinforcement learning, often find very complicated strategies. For example, AI agents can play Go at a super human level (Silver et al. 2017) and can often solve the Rubik's cube in the most efficient way possible (Agostinelli et al. 2019). Explanations for these strategies may not be of use to most people. Therefore, we seek to constrain the space of explanations to human-understandable explanations. While the underlying planner may have a complicated and efficient plan, such plans will only be incorporated if they can be explained. Furthermore, we naturally incorporate compositionality in our approach by representing an explanation as a directed graph, allowing explanations to be re-used. Therefore, if a plan for solving a given state is too difficult to explain, we can then find plans to reach intermediate points in the graph that may be easier to explain, thus favoring comprehensability over optimality.

In our approach, explanations for how to solve all instances of a problem will be derived from a directed acyclic graph whose nodes are first-order logic programs representing sets of states and whose edges represent macro actions. All nodes in this graph will have a path to the goal node. A path to the goal node can be obtained from a given node by applying any macro-action associated with the outgoing macro-actions of that node. Intuitively, each node can be thought of as a precondition for applying one of its outgoing macro-actions. The preconditions will be induced from observed solutions using inductive logic programming (ILP) techniques (Cropper et al. 2022). Given enough observations, the union of the preconditions for each node should cover the entire state space. Therefore, one can then solve any given state by finding a node that entails that state and applying any sequence of macro-actions that leads to the goal node. We refer to this as an explainable pathfinding graph, or e-graph.

Since logic programming is often used for XAI techniques due to its symbolic structure, given well-design predicates and inductive biases, each node can be explained to a human. In this work, we focus on finding the simplest explanations possible. We discuss a variety of ways one can quantify simplicity and how it can be achieved. We seek to describe the entire e-graph to a human in a clear and concise manner, thereby explaining how to solve all instances of the problem. In this work, we investigate how we can do this by learning hierarchical structures of the graph as well as using the graph to learn simple if/else rules.

## Related Work

The need for explanation in planning was articulated initially in (Fox, Long, and Magazzeni 2017; Hoffmann and Magazzeni 2019) and this lead to introduction of a number of approaches. A recent review summarizes the progress and ongoing challenges (Chakraborti, Sreedharan, and Kambhampati 2021). In brief, here the automated explanation method attempts to explain in a situation (state), why the planner chose an action, why the planner did not choose an action, why the planner decisions are better, why things asked (e.g., goals) can not be done, why one needs to replan, and why one does not have to replan.

Defining solutions to sequential decision making problems as programs has been proposed to create explainable agents. Using a given policy as an oracle, search can be done in a program space to match this oracle as close as possible (Verma et al. 2018). A drawback to this method is that the oracle's policy may be very complex, causing the learned program to be hard to understand. A program can be embedded in a latent space and a search can then be done in this latent space to find a program that solves a given problem (Trivedi et al. 2021). One can model constructing a program itself as a reinforcement learning task where each function call is an action (Ellis et al. 2021). This approach allows one to grow a library of functions to augment its action space to solve increasingly complex problems. Though programs have a discrete structure, one can optimize this a relaxed continuous structure instead (Liu, Simonyan, and Yang 2018). This relaxed structure has been used to allow the learning of nested if/then/else programs for solving mazes (Qiu and Zhu 2021).

The e-graph is similar to the planning graph created as part of the Graphplan planner (Blum and Furst 1997) but has notable differences. In Graphplan, the planner first creates a graph, alternating in states and actions levels, starting with the initial state and representing actions that become applicable and states that could be reached, and so on. In Graphplan, the graph is used to search for a plans to solve a particular instance of a problem. In our work, we construct the e-graph that can solve all instances of a problem.

## Background

### Towers of Hanoi (ToH)

The game of Towers of Hanoi is one player game with three pegs and a varying number of disks. The disks are of different sizes and are arranged in ascending order of size. To solve the problem all the disks must be moved to the last peg. This is done by moving one disk at a time so that the disk can be directly kept on a peg or over a larger disk only. The action space we use allows one to move any disk to any peg (in format (*disk, to-peg*)) resulting in nine actions in total. While a typical action space would also require specifying the peg that the disk is moving from, we use this action set as it allows us to learn a smaller graph. Invalid actions result in no change.

## Problem Definition

The pathfinding problem is represented as a Markov decision process (MDP) (Puterman 2014). The MDP can be de-

fined as a set of states $\mathcal{S}$, set of actions $\mathcal{A}$, state-transition probability function $T$, and expected reward function $r$. The state-transition probability function returns the probability of transitioning from some state $s$ to another states $s'$ using action $a$, $T(s, a, s')$. We limit the scope of this work to deterministic environments, therefore, we can replace $T$ with a function $A$ that returns the resulting state $s'$ when taking action $a$ in state $s$, $A(s, a)$. The reward function $r$ returns the expected reward when taking action $a$ in state $s$ and transitioning to state $s'$, $r(s, a, s')$. In the context of pathfinding, this is the negative transition cost.

The objective of pathfinding is to find a path from any given state $s$ to a goal state $g$, where $g$ is in the set of goal states which is a subset of $\mathcal{S}$. Given any planner that is used to solve a pathfinding problem (i.e. based on search, dynamic programming (Bellman 1957), or a combination of the two (Chen and Wei 2011; Agostinelli et al. 2019)), we seek to use this planner to find an explanation of how to find a path from any given state to a goal state. While it is often desirable to find a shortest path, we instead seek to find an explanation that is easiest for humans to understand. In the following section, we will describe how we formally quantify the simplicity of an explanation.

## Approach

### Graph Semantics and Structure

The explainable pathfinding graph, or e-graph, is a graph, $\mathcal{G}$, whose nodes represent sets of states with first-order logic programs and whose edges represent macro-actions, where macro-actions are a sequence of atomic actions. We refer to the first-order logic programs as preconditions, since they must be satisfied in order for the macro-action to be applicable. The graph is directed and acyclic. Every node in the graph has at least one child node except the single terminal node which represents the goal. The graph first starts out with only the goal node. Every node added must have a child node that is already in the graph. Therefore, the graph should be complete, meaning there is a path to the goal from any goal node. Furthermore, given enough observations, the graph should also be sound, meaning it should not be possible to come up with an invalid plan.

The preconditions at each node are constructed from user-defined predicates. When these predicates are learned, the hypothesis space is constrained by a user-defined inductive bias. Given predicates that a user can understand and an inductive bias that prunes complicated programs, every learned precondition should be explainable. The overall explanation is then any method that summarizes the entire graph. This includes a hierarchical representation, a representation that groups similar macro-actions together, or a set of examples whose solutions match the preconditions of every node.

In this work, we concern ourselves only with finding explanations that are easy to understand. To measure this, we define a cost for the e-graph, $c(\mathcal{G})$. This cost can take into consideration the number of predicates, number of nodes, and number of macro actions. Furthermore, it could even be based on the compactness of the final explanation used to

summarize the graph, such as one obtained from natural language generation techniques. In our work, we simply seek to minimize the sum of the cost of the edges in the graph where the cost of an edge is determined by the number of macro actions. Therefore, compositionality will be needed to achieve this. For example, one could have a node with an edge to the goal with a cost of one and another node with an edge to the goal with a cost of two, resulting in a graph with a cost of three. However, if one can compose these nodes using shorter edges, then we can get a graph of a lower cost. A figure illustrating this is shown in Figure 1. We take a greedy approach to this optimization process. However, in future work, we aim to use other optimization approaches and as well as other metrics of cost to get the most human-understandable explanations.

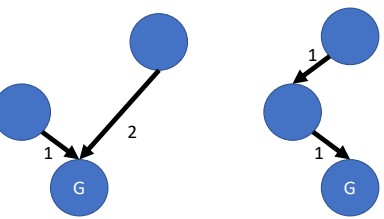

Figure 1: Left: A graph with a cost of three. A graph with a cost of two.

## Learning the Graph

The graph is initialized with a single goal node that represents the set of all solved states. We then initialize a set of states that will be used to generate macro actions. This set of states is obtained by randomly generating states and using a pathfinding algorithm, $\rho$, to find a path from all generated states to a goal state. We then seek to add nodes to the graph until every state in this set is entailed by a node in the graph.

Each node added to the graph must have as its child an existing node in the graph. The edge connecting the two nodes must represent a macro action that can transform any state entailed by the parent's precondition to a state entailed by the child's precondition. To achieve this, we maintain a priority queue that contains tuples of macro-actions and nodes, where each macro-action has successfully transformed some state into a state entailed by its corresponding node. Since we are taking a greedy approach, this priority queue is sorted according to the macro action length. When there is a tie, the closest ancestor of the goal node is given priority.

To obtain these tuples of macro actions and nodes, for a given node, $\rho$ is used to find a path from all unsolved states to a state entailed by that node. The first node for which this is done is for the goal node. However, since the nodes represent sets of states in first order logic, the processes of finding a path to a set of states must be addressed. In our work, we use an underlying Prolog representation to use the first-order logic program as a query and get all states in the set. We then can use a heuristic function to estimate which state is closest and find a path to that state. For every successful path, the macro action and corresponding node are added to the priority queue. However, this approach will not scale as the size

of the set grows. In the discussion section, we will consider other approaches.

When a macro-action and corresponding node is removed from the priority queue, we apply the macro-action to all unsolved states. Unsolved states that the macro-action transforms into states that are entailed the corresponding node's precondition are used as positive examples while the other states are used as negative examples. Furthermore, all solved states (that is, states entailed by a node that is already in the graph) are also used as negative examples. We then seek to learn a first-order logic program that perfectly separates the positive and negative examples. Learning may not always be successful. For our purposes, this is desirable because we only want to add a node to the graph if it can be explained to a human. We use the Popper (Cropper and Morel 2021) inductive logic programming software to learn preconditions. Popper's flexible approach to specifying inductive biases allows us to constrain the hypothesis space for both speed and for comprehensibility. For example, in Towers of Hanoi, including simple constrains, such as specifying that disks can only have one size and that a disk cannot be on more than one peg, result in much faster learning times. Also, including constraints for comprehensibility, such as specifying that the predicate to say a disk is not on a peg can only be used once per clause, results in programs that are much easier to understand.

The overall algorithm is showed in Algorithm 1 and the process to obtain tuples of macro-actions and nodes is shown in Algorithm 2.

## Explanation Generation

Given an e-graph, we explore different approaches for obtaining an explanation. The first is a simple template-based natural language generation approach that explains the preconditions for each node. We can even print the entire graph as a nested structure where ancestors are nested within their descendants. In the future, we plan to utilize language models based on transformers (Vaswani et al. 2017) to convert first-order logic sentences into everyday language.

The next approach is obtaining a hierarchical structure. One can obtain learn hierarchical nodes by first obtaining all states associated with a given node and all states associated with its descendants. Then, one can learn a logic program that entails all of the aforementioned states and none of the states entailed by any other nodes in the graph. We explore this approach by first trying to learn programs for nodes for the greatest number of descendants. If the program is successful, we then remove the node from the descendants of all other nodes. We then repeat this processes until all nodes are in a hierarchical node, even if the hierarchical node just contains a single node. We can also do the same thing for the ancestors of nodes.

The final approach is to group nodes together if their outgoing edge has the same macro-action. Then, we can learn a logic program that entails all states entailed by these nodes and none of the states entailed by the other nodes. The intuition is that we can find a simple if/then structure for when to apply macro-actions. If there are not many macro-actions,

**Algorithm 1:** e-graph Construction

**Input:**
    $\mathcal{S}_u$: Set of unsolved states
    $P$: Goal precondition
    $B$: Background knowledge
    $\rho$: Pathfinding strategy
**Output:**
    $g$: Goal node of e-graph
$g = Node(P)$
$\mathcal{S}_s = \{\}$ //Set of solved states
Remove states entailed by $P$ from $\mathcal{S}_u$ and add to $S_s$
$q = []$ //Priority queue of macro actions and nodes
add_macros($\mathcal{S}_u, g, q, \rho$)
**while** *len($\mathcal{S}_u$)> 0* **do**
    $m, n = q.pop()$ //macro action and target node
    $E^+ = []$ //positive examples
    $E^- = \mathcal{S}_s$ //negative examples
    **for** $s \in \mathcal{S}$ **do**
        Apply $m$ to $s$ to obtain $s'$
        // check if result is entailed by node
        **if** $n.P(s')$ *is True* **then**
            $\llcorner$ append $s$ to $E^+$
        **else**
            $\llcorner$ append $s$ to $E^-$
    $P$ = learn_precondtion($E^+, E^-, B$)
    **if** *P learned successfully* **then**
        $n = Node(P)$
        Remove states entailed by $P$ from $\mathcal{S}_u$ and add
          to $S_s$
        $\llcorner$ add_macros($\mathcal{S}_u, n, q, \rho$)
**Return** $g$

---

**Algorithm 2:** Add Macro Actions

**Input:**
    $\mathcal{S}$: Set of states
    $n$: Node
    $q$: Priority queue
    $\rho$: Pathfinding strategy
$\mathcal{M} = \{\}$ //set of macro actions
**for** $s \in \mathcal{S}$ **do**
    //$m$ is macro action that takes $s$ to a state entailed
      by the precondition of $n$.
    $m = \rho(s, g)$
    **if** $m$ *found successfully* **then**
        $\llcorner$ add $m$ to $\mathcal{M}$
**for** $m \in \mathcal{M}$ **do**
    add $(m, n)$ to $q$ with first priority len($m$) and
    second priority being the depth of $n$

---

then someone can read through each one to understand the explanation.

## Results

The background knowledge we are using for Towers of Hanoi contains predicates that describe a disk being on a peg, a disk not being on a peg, a disk being above another disk, a disk's top or bottom being clear, and a peg being clear. Since we are only considering the three disk scenario, we use breadth-first search (BFS) for finding macro actions. BFS is one of the simplest path finding algorithm in which the shallowest nodes are given priority. The algorithm is optimal assuming the transition costs between each node is one. Towers of Hanoi has a state space with 27 states.

The learned e-graph is shown in Figure 2a. We use a nested structure to represent the graph which has 15 nodes. A template-based approach was used to generate a natural language description of each node. A total of seven unique macro actions are used in the solution and are omitted in the figure for brevity. In this case, the all macro-actions are one of the nine atomic actions. To simplify this explanation, we look at how we can obtain hierarchies. The first approach to create a hierarchical representation is to learn programs

to differentiate a node and all its descendants from all other nodes. The results of this approach are shown in Figure 2b. This results in a graph with 12 nodes. We then create a hierarchical by learning programs to differentiate a node and all its ancestors from all other nodes. The results of this approach are shown in Figure 2c. The results show a very simple representation that says that the largest disk goes to peg3 from either peg1 or peg2.

Finally, we investigate learning an if/else structure by grouping nodes with the same macro-action together. Since there are seven macro-actions used, we should have seven if/else statements. While the ILP system was not able to learn a program for moving disk1 to peg1 or moving disk1 to peg2, it successfully learned a precondition for the other five macro actions. The result is shown in Figure 2d.

## Discussion

### Applications to the Rubik's Cube

We are currently extending this work to the Rubik's cube. The Rubik's cube poses unique challenges for any XAI approach due to its symmetries. Furthermore, the fact that solutions to the Rubik's cube can be explained to young children sets a standard that current XAI approaches based on modern AI techniques, such as deep learning, cannot match. In our preliminary work, we using an example based explanation approach that uses landmarks for clarity. That is, after the graph is constructed, we find a set of example states where each node entails at least one state in the union of states in the solution paths. We identify specific landmarks based on the number and length of incoming edges to a node. We then ensure we explicitly explain these landmarks. On the other hand, we implicitly explain other nodes that only have one parent or whose incoming edges are short. This way, we can try to avoid obvious explanations while focusing on more difficult aspects of the problem.

For finding macro-actions using a search strategy $\rho$, we extend the DeepCubeA algorithm (Agostinelli et al. 2019). In this domain, however, the preconditions for nodes often only contain a subset of the stickers on the Rubik's cube.

```
state is goal
      disk2 is above disk3, disk2 is on peg3, disk2's top is clear
            disk3's top is clear, disk3 is on peg3, disk1's bottom is clear
                  disk2's bottom is clear, disk2 is on peg1, peg2 is clear
                        disk3's top is clear, disk3 is on peg2, peg3 is clear
                              disk2 is on peg1, disk2's top is clear, disk3 is on peg2
                                    disk3 is on peg2, disk2's top is clear, peg1 is clear
                                          disk2 is above disk3, disk2 is on peg2, peg3 is clear
                                          disk2 is on peg3, disk3 is on peg2, disk3's top is clear
                  disk2 is on peg2, disk2's bottom is clear, peg1 is clear
                        disk3's top is clear, disk3 is on peg1, peg3 is clear
                              disk2 is on peg2, disk2's top is clear, disk3 is on peg1
                                    disk2's top is clear, disk3 is on peg1, peg2 is clear
                                          disk3 is on peg1, disk2 is on peg1, peg3 is clear
                                          disk3 is on peg1, disk3's top is clear, disk2 is on peg3
```

(a) A visualization of the learned graph. There are a total of 15 nodes.

```
state is goal
      disk3 is on peg3, disk1's bottom is clear
            disk2 is on peg2, disk2's bottom is clear, peg1 is clear
                  disk2 is on peg2, disk3 is on peg1
                        disk2's top is clear, disk3 is on peg1, peg2 is clear
                              disk3 is on peg1, disk3's top is clear, disk2 is on peg3
                              disk3 is on peg1, disk2 is on peg1, peg3 is clear
            disk2's bottom is clear, disk2 is on peg1, peg2 is clear
                  disk2 is on peg1, disk3 is on peg2
                        disk3 is on peg2, disk2's top is clear, peg1 is clear
                              disk2 is on peg3, disk3 is on peg2, disk3's top is clear
                              disk2 is above disk3, disk2 is on peg2, peg3 is clear
```

(b) The hierarchical graph obtained by creating hierarchical nodes using node descendants. There are a total of 12 nodes.

```
disk3 is on peg3
      disk3 is on peg2
      disk3 is on peg1
```

(c) The hierarchical graph obtained by creating hierarchical nodes using node ancestors. There are a total of 3 nodes. This recovers a natural hierarchical structure associated with Towers of Hanoi.

```
IF disk2 is above disk3, disk1 is not on peg3 THEN move disk1 to peg3
IF disk3 is on peg3, disk3's top is clear, disk1's bottom is clear THEN move disk2 to peg3
IF disk3's top is clear, peg3 is clear THEN move disk3 to peg3
IF disk2's top is clear, disk3 is not on peg3, peg1 is clear THEN move disk2 to peg1
IF disk3 is not on peg3, disk2's top is clear, peg2 is clear THEN move disk2 to peg2
```

(d) An if/else structure obtained by grouping nodes with the same macro-action together. There were a total of seven macro actions. The learning was successful for all macro-actions except two.

Figure 2: Four different explanation methods. Actions used to connect edges have been omitted for brevity.

Therefore, there are many elements of the state that are not specified. To allow the neural network to find paths to partially specified states, we use hindsight experience replay (Andrychowicz et al. 2017) and a special color for non-specified stickers, to learn a cost-to-go function that estimates the distance between any pair of partially specified states.

## Searching in the Space of States vs Sets of States

Currently, we are finding macro-actions that can transform a state to another state entailed by the precondition of a node by generating states with Prolog and finding paths in the state space. However, this process will become unsustainable as the number of states entailed by each node grows. One way to try to address this is by using subgoals where states are only partially specified. The state space of a partially specified subgoal can be much smaller than the overall state space. Such an approach is often seen when describing solutions to the Rubik's cube. If this is not possible, then one

will have to be able to search in the space of sets of states instead of the space of states. One possible approach to accomplish this is to use hindsight experience replay (Andrychowicz et al. 2017) to learn a cost-to-go function that takes as input a given state and a logic program. The cost-to-go function should estimate the distance between a given state and the closest state entailed by the logic program. One could generate training examples by first generating a goal state in the state space and then creating a first-order logic program that entails that goal state.

## Abstraction While Learning the Graph

Our current explanation methods are able to build abstractions by learning a hierarchical graph after the e-graph has been learned. However, abstraction may be needed during the learning process, itself. For example, learning to transition to a specific node may be too cumbersome for the ILP system, especially if many variables are involved due to interdependence. Instead, one could relax the exact transition

that should take place by allowing it to transition to a set of nodes.

## Explanation Aware Search

Part of the ease of an explanation involves how easy it is to describe the macro action. For example, there a longer macro-action composed of only one atomic action may be easier to understand than a shorter macro-action composed of many atomic actions. Furthermore, atomic-actions can be represented as a predicate with arguments. For example, in Towers of Hanoi, the disk and destination peg can be the arguments to the action predicate. In this formulation, the precondition and macro-action can share variables and create more compact descriptions of macro-actions. It is possible that this work could be combined with methods based on learning with deep neural networks, such as Dreamcoder (Ellis et al. 2021) or DeepCubeA (Agostinelli et al. 2019) to exploit this relationship.

## Acknowledgments

This work was funded by the University of South Carolina ASPIRE-II program.

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
