# OpenReview forum: "Explainable Pathfinding for Inscrutable Planners with Inductive Logic Programming"
_icaps-conference.org/ICAPS/2022/Workshop/XAIP — XAIP 2022_

### Official Review · Reviewer_8Czq · 2022-04-26
**An inductive logic programming-based approach to generating interpretable policies for inscrutable planners.**

**Rating:** 7
**Confidence:** 3

**Review:**

This paper takes an inductive logic programming approach to learning so-called explainable pathfinding graphs (e-graphs) based on observations of solutions generated by planners whose solutions may otherwise be inscrutable to humans. The paper proposes a number of ways in which e-graphs may be ‘parsed’ to generate explanations for human users of such systems. The ILP method used can incorporate background knowledge and constrained such that only ’simple enough’ e-graphs are generated. The paper uses Towers of Hanoi as a running example and even discusses application potential to the Rubik’s cube domain.

-typos-
* Related work section
    * to introduction of a number of approaches -> to **the** introduction of a number of approaches
    * Though programs have a discrete structure, one can optimize this a relaxed continuous structure instead  -> remove ‘this' or add ‘using’ before ‘a’
    * In Graphplan, the graph is used to search for a plans -> change ‘plans’ to ‘plan'
    * In our work, we construct the e-graph that can solve all instances of a problem. -> change ’the e-graph’ to ‘an e-graph'
* background
    * The game of Towers of Hanoi is one player game -> The game of Towers of Hanoi is **a** one player game
    * transitioning from some state s to another states s' -> change 'another states' to 'another state'
* Learning the graph
    * are entailed the corresponding node’s precondition are used as positive examples -> are entailed **by** the corresponding node’s precondition are used as positive examples
* Results
    * We then create a hierarchical by learning programs -> We then create a hierarchical *graph* by learning programs
* Discussion
    * In our preliminary work, we using an example based -> In our preliminary work, we *use* an example based
    * For example, there a longer macro-action composed of only one atomic action -> For example, a longer macro-action composed of only one atomic action

Suggestions and Questions to the Authors
* In the related work section it is mentioned that “… here the automated explanation method attempts to explain in a situation (state), why the planner chose an action, why the planner did not choose an action, why the planner decisions are better, why things asked (e.g., goals) can not be done, why one needs to replan, and why one does not have to replan.” Upon first reading this section, I was anticipating that the approach described in the paper would be able to handle some of these explanation types. However, it is my understanding that the explanations derived from e-graphs can ‘only’ make interpretable the policy for all instances of some task. I wonder if the authors could discuss how e-graphs can be used to generate, for example, contrastive explanations and other types of explanations mentioned in the related work section.
* Related to the previous point, it would be interesting to consider the relation of explanations derived from e-graphs to work on model reconciliation [1] which aims to resolve misconceptions held by the recipient of an explanation (typically a human user of a planning system). For example, in the Rubik’s cube domain, a human observer may not be aware of some of the dynamics of the cube and may therefore be confused by, say, some of the if-then rules generated by the approach described in the paper.
* I would have liked to see a more detailed comparison to Sreedharan et al.’s (2022) work mentioned in the related work section. The paper mentions that this work focuses on solving only one instance and that the current paper in contrast can solve all instances of a problem. Could the authors discuss how their work compares to Sreedharan et al.’s when given a single instance?
* I appreciated that the paper discussed the ‘simplicity’ of explanations to a human and the quantification of this notion using the cost of e-graphs. While the following consideration is likely beyond the scope of the current paper, I was nevertheless wondering if the authors have given thought to the question of the *perceived* simplicity and interpretability of the various forms of explanations derived from e-graphs. For instance, are such explanations meant mostly for experts with intimate knowledge of the domain dynamics or are they also intended for novice users?
* Since the term ‘explanation' is used somewhat differently in different places in the paper (e.g., "by representing an explanation as a directed graph” in the intro and then "The overall explanation is then any method that summarizes the entire graph” later in the paper) , it would be helpful to have a (at least somewhat) formal definition of an explanation.
* The use of ‘pathfinding’ in the paper makes intuitive sense. However, it is a charged term in the planning community due to the rich literature on path planning. I was therefore wondering why it was chosen and whether ‘explainable planning’ would also accurately describe the problem addressed in the paper.
* As a reader not overly familiar with inductive logic programming, I found reading the technical section of the paper challenging without going to the literature. I think it would be helpful to readers if the paper included concrete pointers to ILP exposition and possibly, if there’s space, an ILP subsection in the background section.

[1] Chakraborti, T.; Sreedharan, S.; Zhang, Y.; and Kambham- pati, S. 2017. Plan Explanations as Model Reconciliation: Moving Beyond Explanation as Soliloquy. In IJCAI, 156– 163.

---

### Official Review · Reviewer_PV2U · 2022-04-26
**Interesting idea, user input not clear, odd to name it explainable "pathfinding"**

**Rating:** 7
**Confidence:** 4

**Review:**

I find the idea of explaining the space of plans to the goal to be interesting, and I believe using ILP for that is novel too.
The paper is mostly well written, though it could benefit from a short conclusion section at the end (and a comment regarding the need to evaluate this kind of explanation with users).

Pros:
- Interesting idea and method
- Explanations seem human readable/intuitive (to me)

Cons:
- It is not clear to me what input users (or any human) actually need to give to the method, together with the problem description, in order for explanations to be generated. The text suggests users define predicates / an inductive bias. Maybe I missed this but what would this be in Tower of Hanoi case?
- For me it is odd to name the method an explainable "pathfinding" method. Typically the term "pathfinding" is used for very spatial applications like computer games and robotics - but the method here seems very general and detached from spatial meaning. The method does not automatically ground the explanations using any spatial knowledge so I find it hard to imagine how one would use it in real-world pathfinding (games/robots). Some methods focusing on explainable pathfinding specifically actually do scale to these settings (e.g. Brandao et al "Explaining Path Plan Optimality: Fast Explanation Methods for Navigation Meshes using...", ICAPS2021). Why not call it explainable planning instead, or explaining the "space of plans" (e.g. Eifler et al "Plan-Space Explanation via Plan-Property Dependencies", IJCAI2020).
- No user study

Minor issues:
- Please (at least informally) define what "entailing" means ("a node that entails a state"; "entailed by a precondition")
- In the introduction, the authors say "we focus on finding an explanation for solving all possible instances of a problem" but do not give a motivation for why this would be a good/useful/important thing to do.
- This seems strange to me: "there is a path to the goal from any goal node"
- In this sentence:  "Unsolved states that the macro-action transforms... negative examples." there is a problem with punctuation, and maybe "by" is missing after "entailed"
- In this sentence: "The results show a very simple representation that says that the largest disk goes to peg3 from either peg1 or peg2." ... actually, the explanation is not written in this way, and authors had to interpret the explanation to write it in this way (even if trivially). This form could have an impact in terms of how "simple" or intuitive the explanation is for users (requires user study)
- Please discuss the issues behind (and consequences for explainable systems given) the lack of "if" statements learned by the ILP system.
- "we using"
- "there a longer"

---

### Meta-Review · Program_Chairs · 2022-04-30

**Recommendation:** Accept
**Confidence:** 5

**Metareview:**

Both reviewers agree that the paper presents a novel solution (i.e., the use of ILP-based techniques) to a very relevant problem (i.e. explaining a space of plans). The reviewer does point to some clarity issues, in particular inconsistent or incongruent use of some terms in particular the term 'pathfinding'. I hope the authors will get a chance to address these problems in the future draft. Also going forward the authors would also consider running user studies to verify their method, a recommendation also made by Reviewer PV2U. All in all I would recommend the acceptance of the paper.

---

### Decision · Program_Chairs · 2022-04-30

Accept